# Dendritic small conductance calcium-activated potassium channels activated by action potentials suppress EPSPs and gate spike-timing dependent synaptic plasticity

**Scott L Jones[1], Minh-Son To[1,2], Greg J Stuart[1]\***

[1]Eccles Institute of Neuroscience and Australian Research Council Centre of Excellence for Integrative Brain Function, John Curtin School of Medical Research, Australian National University, Canberra, Australia; [2]Department of Human Physiology and Centre for Neuroscience, Flinders University, Adelaide, Australia

**Abstract** Small conductance calcium-activated potassium channels (SK channels) are present in spines and can be activated by backpropagating action potentials (APs). This suggests they may play a critical role in spike-timing dependent synaptic plasticity (STDP). Consistent with this idea, EPSPs in both cortical and hippocampal pyramidal neurons were suppressed by preceding APs in an SK-dependent manner. In cortical pyramidal neurons EPSP suppression by preceding APs depended on their precise timing as well as the distance of activated synapses from the soma, was dendritic in origin, and involved SK-dependent suppression of NMDA receptor activation. As a result SK channel activation by backpropagating APs gated STDP induction during low-frequency AP-EPSP pairing, with both LTP and LTD absent under control conditions but present after SK channel block. These findings indicate that activation of SK channels in spines by backpropagating APs plays a key role in regulating both EPSP amplitude and STDP induction.
DOI: https://doi.org/10.7554/eLife.30333.001

**\*For correspondence:**
Greg.Stuart@anu.edu.au

**Competing interests:** The authors declare that no competing interests exist.

## Introduction

Small conductance calcium-activated potassium channels (SK channels) in spines regulate the amplitude of EPSPs in pyramidal neurons in the hippocampus and amygdala (*Bloodgood and Sabatini, 2007*; *Faber et al., 2005*; *Ngo-Anh et al., 2005*). This earlier work has shown that calcium influx into spines during EPSPs leads to SK channel activation, which acts to reduce NMDA receptor (NMDAR) activation presumably by promoting voltage-dependent magnesium block (*Mayer et al., 1984*; *Nowak et al., 1984*). These studies indicate that during EPSPs in pyramidal neurons in the hippocampus and amygdala SK channels in spines form a negative feedback loop controlling EPSP amplitude and calcium influx (*Bloodgood and Sabatini, 2007*; *Faber et al., 2005*; *Ngo-Anh et al., 2005*). In addition to their impact on EPSPs, we recently showed in cortical pyramidal neurons that SK channels in spines also act to limit calcium influx into spines and dendrites during backpropagating APs (*Jones and Stuart, 2013*).

   Given the central role of NMDAR activation and calcium in synaptic plasticity (*Bliss and Collingridge, 1993*; *Yang et al., 1999*), the capacity of SK channels to influence NMDAR activation and calcium influx into spines during EPSPs and APs is likely to have consequences on the induction of synaptic plasticity. Consistent with this idea, blocking SK channel activation, and thereby the impact of SK channels on NMDAR function, can promote and enhance long-term potentiation (LTP) evoked

by tetanic stimulation in pyramidal neurons from the hippocampus (*Behnisch and Reymann, 1998*; *Stackman et al., 2002*) and amygdala (*Faber et al., 2005*). Loss of SK channels from spines has also been shown to contribute to LTP expression in hippocampal pyramidal neurons (*Lin et al., 2008*). Finally, a recent study observed that mGluR activation facilitates LTP induction during STDP in hippocampal neurons via inhibition of SK channels (*Tigaret et al., 2016*).

Here we investigate the impact of SK channels activated by backpropagating APs on subsequently evoked EPSPs as well as the induction of STDP. We find that EPSPs in both cortical and hippocampal pyramidal neurons are suppressed by preceding APs if these APs are evoked in a brief time window before EPSP onset. EPSP suppression by preceding APs required activation of dendritic SK channels, with the magnitude of EPSP suppression dependent on the distance of activated synapses from the soma. Furthermore, we show the extent of EPSP suppression was NMDAR dependent. Finally, we show that SK channel activation by APs acts to gate the induction of STDP during low-frequency single AP-EPSP pairing.

## Results

### Effect of SK channel activation on EPSPs paired with APs

To investigate the acute impact of SK channel activation by backpropagating APs on subsequently evoked EPSPs, unitary EPSPs were evoked with and without preceding postsynaptic APs during recordings from synaptically connected layer 5 pyramidal neurons (*Figure 1a*). Postsynaptic APs generated on their own were interleaved between trials and digitally subtracted from the AP-EPSP waveform to reveal the impact of preceding postsynaptic APs on unitary EPSPs. Generation of single APs 10 ms before EPSPs (−10 ms timing) suppressed EPSP peak by approximately 30% (*Figure 1b, c*; n = 6). This EPSP suppression was significantly reduced by bath application of the SK channel blocker apamin, indicating a role of SK channels (*Figure 1b,c*). The magnitude of SK-dependent EPSP suppression was dependent on the precise timing of EPSPs and postsynaptic APs, with EPSP suppression only observed when postsynaptic APs were evoked within a brief time window before EPSP onset (*Figure 1d,e*; n = 6). EPSP width was also reduced by preceding postsynaptic APs in a time and SK-dependent manner (*Figure 1f*). EPSP suppression by preceding APs was not totally abolished by bath applications of apamin (*Figure 1e*), indicating that in addition to SK channels other postsynaptic conductances activated by APs contribute to suppression of subsequently evoked EPSPs. In summary, these results indicate that, together with other conductances, SK channels activated by postsynaptic APs significantly reduce subsequently evoked EPSPs in time-dependent manner.

Previous findings in basal dendrites of layer 5 pyramidal neurons indicate that calcium influx into dendritic spines during backpropagating APs is regulated by SK channels, with the impact of SK channels on spine calcium influx greater for spines located further from the soma (*Jones and Stuart, 2013*). We therefore investigated the dependence of EPSP suppression by APs on synapse location. To do this, we used somatic EPSP rise-time as a proxy for synapse location. Previous work indicates that somatic EPSP rise time is positively correlated with the distance of active synapses from the soma of layer 5 pyramidal neurons (*Letzkus et al., 2006*), as expected due to dendritic filtering (*Jack et al., 1971*). The range of EPSP rise times in our experiments was similar to that observed in a previous study using paired recordings from layer 5 pyramidal neurons in somatosensory cortex (*Markram et al., 1997a*), with this earlier work finding that synaptic connections between layer 5 pyramidal neurons are made between 80 and 585 μm from the soma (mean: 147 μm; median: 105 μm), with the majority of inputs on basal dendrites. Consistent with our earlier study showing that the impact of SK channels on calcium influx during backpropagating APs is greater at more distal dendritic locations, we found that the impact of SK channel block on EPSP suppression by preceding APs (−10 ms timing) was positively correlated with EPSP rise-time (*Figure 1g*; n = 16). These data suggest that SK channels activated by backpropagating APs lead to greater suppression of EPSPs when they are generated at more distal dendritic locations.

To determine if AP-induced SK channel-dependent suppression of EPSPs also occurs in other neuronal cell types and brain regions, we investigated if APs could suppress EPSPs in cortical layer 2/3 and hippocampal CA1 pyramidal neurons. These experiments indicated that preceding postsynaptic APs suppressed EPSPs in both cortical layer 2/3 and hippocampal CA1 pyramidal neurons in

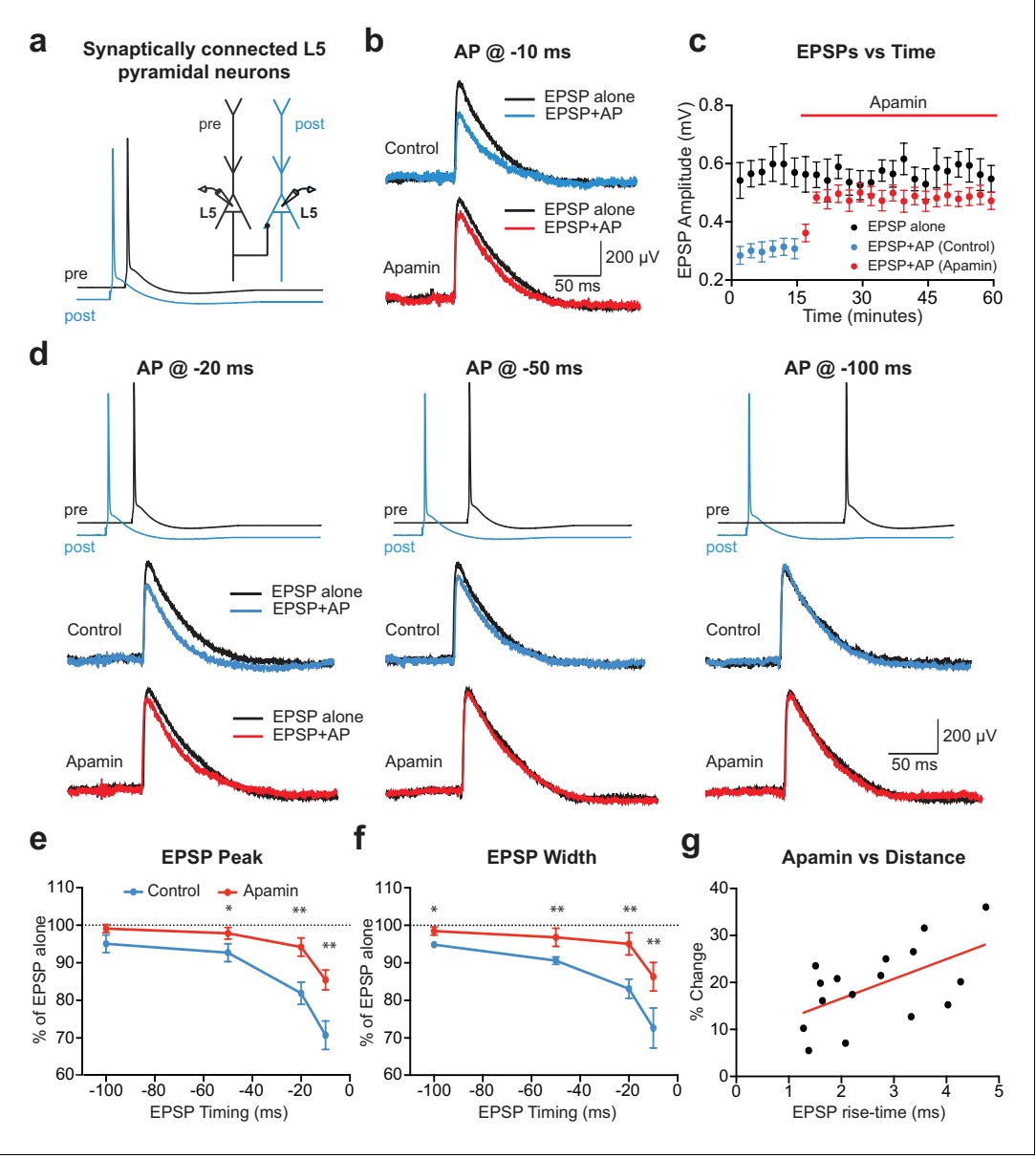

**Figure 1.** APs in cortical layer 5 (L5) pyramidal neurons suppress EPSPs in an SK-dependent manner. (**a**) Experimental arrangement: Simultaneous whole-cell recording from synaptically connected L5 pyramidal neurons. APs in the presynaptic neuron (black) were evoked 10 ms after the AP in the postsynaptic neuron (blue; −10 ms timing). (**b**) Unitary EPSPs evoked alone (black) and 10 ms after APs (blue and red) in control (top) and apamin (bottom). (**c**) Plot of EPSP amplitude versus time for EPSPs evoked alone (black) and 10 ms after APs in control (blue) and apamin (red; same neuron as in b). Each point represents the average response over 2.5 min ± SEM. (**d**) Top: Pre- and postsynaptic APs at −20 ms, −50 ms and −100 ms. Bottom: Unitary EPSPs evoked alone (black) and at different times after APs (blue and red) in the absence (top) and presence of apamin (bottom). (**e,f**) Average EPSP amplitude (**e**) and width (**f**) for unitary EPSPs evoked at different times after APs relative to the amplitude of EPSPs alone in control (blue) and apamin (red; n = 6). (**g**) Percentage increase in unitary EPSP amplitude after apamin for unitary EPSPs evoked 10 ms after APs versus the 10–90% rise time of EPSPs alone (n = 16). Line represents linear fit (R squared = 0.32, p=0.023).

DOI: https://doi.org/10.7554/eLife.30333.002

an SK-dependent manner (*Figure 2a-d*), as observed in layer 5 pyramidal neurons. Interestingly, these experiments also revealed that blocking SK channels had no impact on the amplitude of EPSPs alone in layer 2/3 pyramidal neurons (*Figure 2e*), while it increased the amplitude of

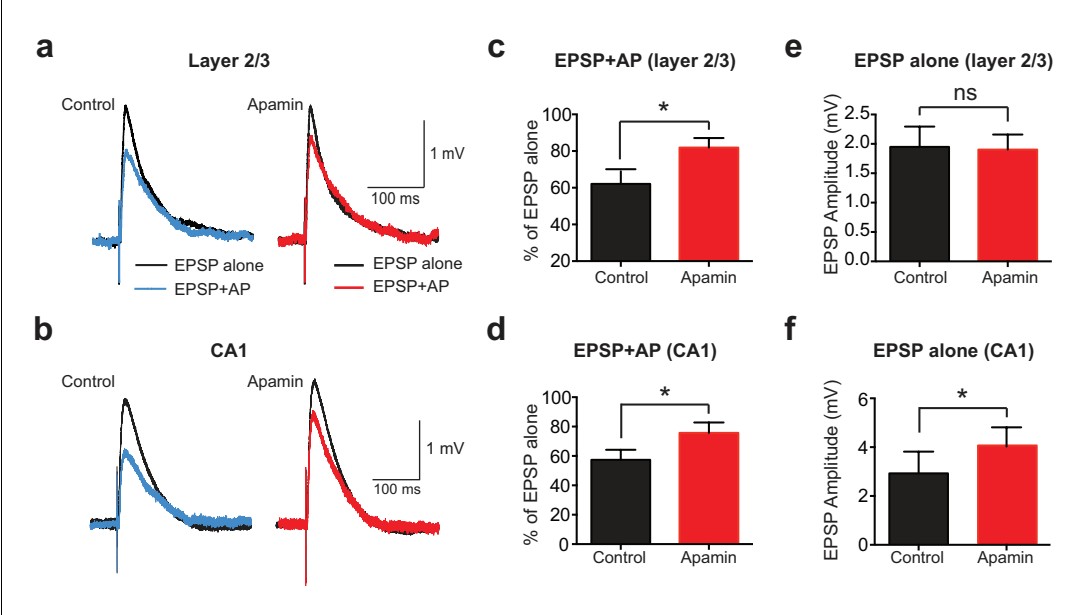

**Figure 2.** APs suppress EPSPs in an SK-dependent manner in cortical layer 2/3 and hippocampal CA1 pyramidal neurons. (a,b) EPSPs evoked alone (black) and 10 ms after APs (blue and red) in control (left) and apamin (right) in layer 2/3 (a) and CA1 pyramidal neurons (b). (c,d) Percentage change in EPSP amplitude for EPSPs evoked 10 ms after APs relative to EPSPs alone in control (black) and apamin (red) in layer 2/3 (c) and CA1 pyramidal neurons (d; n = 7).(e,f) Average EPSP alone amplitude in control (black) and apamin (red) in layer 2/3 (e) and CA1 pyramidal neurons (f; n = 7).

DOI: https://doi.org/10.7554/eLife.30333.003

EPSPs evoked on their own in CA1 pyramidal neurons (*Figure 2f*), as observed by others (*Bloodgood and Sabatini, 2007*; *Ngo-Anh et al., 2005*; *Wang et al., 2014*). Blocking SK channels also had no impact on unitary EPSPs in layer 5 pyramidal neurons (see *Figure 1c*). These observations suggest that, in contrast to CA1 pyramidal neurons (*Wang et al., 2014*), calcium influx via NMDARs is not sufficient to activate SK channels in cortical layer 2/3 and layer 5 pyramidal neurons, and that this requires AP generation. In summary, these data show that preceding postsynaptic APs reduce the amplitude and duration of EPSPs in both cortical and hippocampal pyramidal neurons, with the extent of EPSP suppression dependent on SK channel activation, AP timing and EPSP location.

## Site of SK channel suppression of EPSPs

The finding that SK-dependent suppression of EPSPs by preceding APs is greater for EPSPs generated at more distal dendritic locations (*Figure 1g*) suggests that the SK channels involved are dendritic and presumably activated by backpropagating APs. To test if somatic SK channels also contribute to EPSP suppression we investigated AP-induced suppression of EPSP-like waveforms generated by current injection via the somatic recording pipette. These experiments were performed in layer 5 pyramidal neurons with somatic artificial EPSPs (aEPSPs) evoked alone or with preceding APs at various timing intervals. As the origin of aEPSPs is somatic, AP-induced suppression would be expected to be primarily via mechanisms close to the soma. Artificial EPSPs of 1 mV amplitude were used in these experiments to mimic the average unitary EPSP amplitude observed in layer 5 paired recordings (0.82 ± 0.08 mV; n = 28).

Preceding APs suppressed aEPSPs during pairing at −10 ms, however, the extent of AP-induced suppression of aEPSP peak was much smaller than that observed during unitary EPSPs and was insensitive to apamin (*Figure 3a,b*). An effect of apamin on AP-induced suppression of aEPSP half-width was observed, although this effect was small (*Figure 3c*; n = 7). These data suggest that somatic SK channels activated by APs have little impact on EPSP peak, supporting the idea that AP-induced suppression of unitary EPSPs is primarily due to dendritic SK channels.

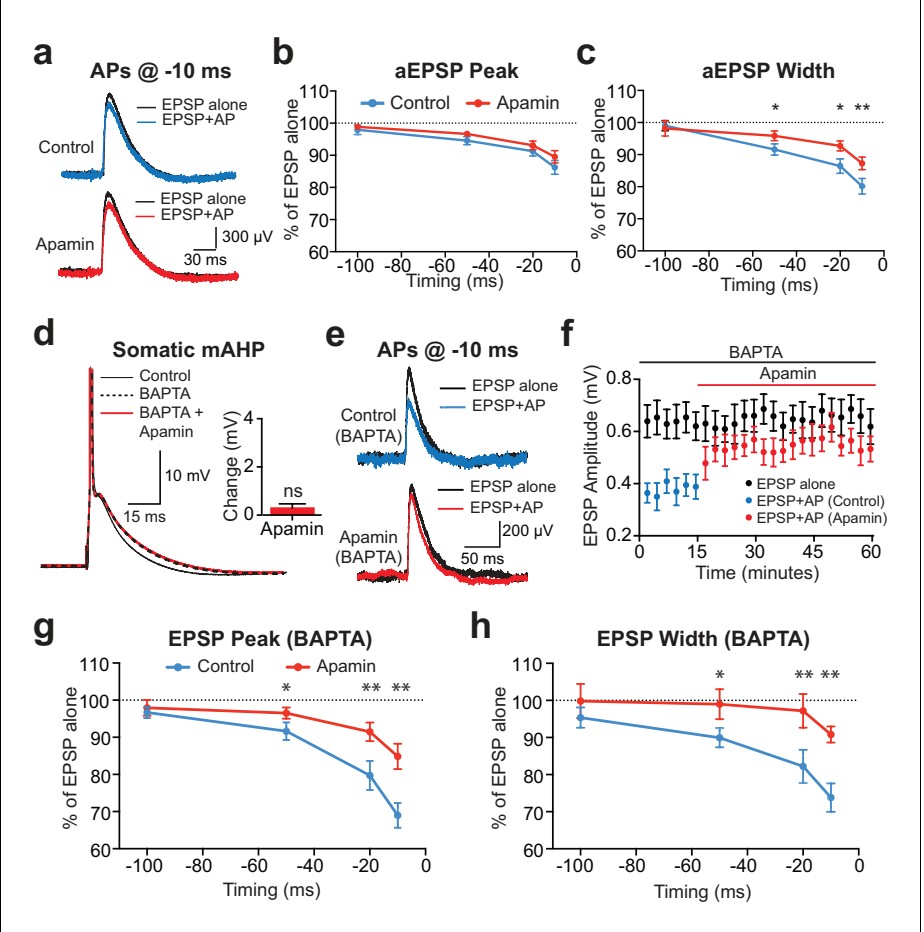

**Figure 3.** Contribution of somatic versus dendritic SK channels to EPSP suppression. (**a**) Artificial EPSPs (aEPSPs) generated by somatic current injection alone (black) and 10 ms after APs (blue and red) in control (top) and apamin (bottom). (**b–c**) Percentage reduction in EPSP amplitude (**b**) and width at 25% of peak (**c**) for somatic aEPSPs evoked at different times after APs relative to the amplitude of aEPSPs alone in control (blue) and apamin (red; n = 7). (**d**) Postsynaptic AP in control (black, solid) and with 300 μM BAPTA in the pipette solution (dashed) before and after addition of apamin (red). Inset: Change in mAHP after apamin with intracellular BAPTA. (**e**) Unitary EPSPs evoked alone (black) and 10 ms after APs (blue and red) in control (top) and apamin (bottom) in recordings with intracellular BAPTA. (**f**) Plot of EPSP amplitude versus time in recordings with intracellular BAPTA for EPSPs alone (black) and 10 ms after APs in control (blue) and apamin (red; same neuron as in e). Each point represents the average response over 2.5 min ± SEM. (**g,h**) Average EPSP amplitude (**g**) and width (**h**) for unitary EPSPs evoked at different times after APs relative to the amplitude of EPSPs alone in control (blue) and apamin (red; n = 6).

DOI: https://doi.org/10.7554/eLife.30333.004

To further dissociate the impact of somatic versus dendritic SK channels on EPSP suppression by preceding APs a low concentration of the calcium chelator BAPTA (300 μM) was added to the somatic pipette solution. We have recently shown that this blocks somatic SK channels in layer 5 pyramidal neurons, while preserving the function of SK channels in dendrites (*Jones and Stuart, 2013*). This effect is not due to poor diffusion of the calcium chelator to dendritic locations, but is likely to result from differences in the localisation of dendritic compared to somatic SK channels with their calcium source (*Jones and Stuart, 2013*). Consistent with these earlier observations, low concentrations of BAPTA completely abolished the SK-mediated component of the somatic mAHP in layer 5 pyramidal neurons (*Figure 3d*). Under these conditions preceding APs suppressed unitary EPSPs to a similar extent to that observed in the absence of BAPTA in an apamin-dependent manner (*Figure 3e–h*; n = 6; compare with *Figure 1*). Together, these data provide strong support

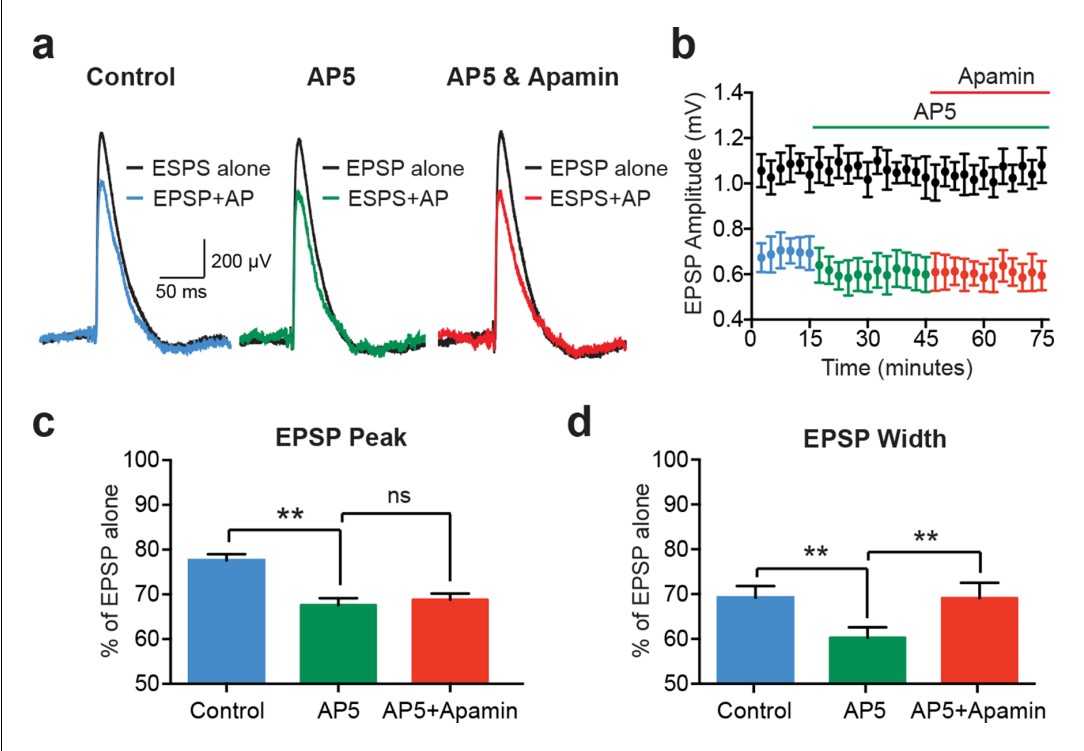

**Figure 4.** EPSP suppression by APs requires NMDARs. (**a**) Unitary EPSPs evoked alone (black) and 10 ms after APs in control (left; blue), AP5 (middle; green) and AP5 plus apamin (right; red). (**b**) Plot of EPSP amplitude versus time for EPSPs alone (black) and 10 ms after APs in control (blue), AP5 (green) and AP5 plus apamin (red; same neuron as in a). Each point represents the average response over 2.5 min ± SEM. (**c,d**) Average EPSP amplitude (c) and width (d) for unitary EPSPs evoked 10 ms after APs relative to the amplitude of EPSPs alone in control (blue), AP5 (green) and AP5 plus apamin (red; n = 5).

DOI: https://doi.org/10.7554/eLife.30333.005

for the idea that dendritic SK channels activated by backpropagating APs are responsible for suppression of subsequently evoked unitary EPSPs.

## Role of NMDA receptors in SK-dependent EPSP suppression

While EPSPs evoked on their own were not affected by SK channel block in cortical pyramidal neurons (*Figure 1c* and *Figure 2e*), previous work in other cell types indicates that SK channels activated during EPSPs suppress NMDAR activation (*Bloodgood and Sabatini, 2007*; *Faber et al., 2005*; *Ngo-Anh et al., 2005*). We therefore investigated the involvement of NMDARs in AP-induced suppression of unitary EPSPs in layer 5 pyramidal neurons (*Figure 4*). Consistent with a critical role of NMDARs, apamin had no impact on EPSP suppression by preceding APs (−10 ms timing) in the presence of the NMDAR antagonist AP5 (*Figure 4a–c*; n = 5). Surprisingly, these experiments revealed that blocking NMDARs with AP5 enhanced EPSP suppression by preceding APs (*Figure 4a–c*; n = 5). Block of NMDARs with AP5 also increased the impact of preceding APs on EPSP width (*Figure 4d*; n = 5). In contrast to observations on EPSP peak, however, in the presence of AP5 EPSP width was slightly increased by SK channel block (*Figure 4d*; n = 5). This small effect on EPSP width is presumably due to block of somatic SK channels, and resembles the impact of SK channel block on AP-induced suppression of somatic aEPSP width (see *Figure 3c*).

## Simulations of SK-dependent EPSP suppression

One interpretation of these observations is that backpropagating APs enhance NMDAR activation, with the extent of NMDAR activation regulated by SK channels. As a result, blocking NMDARs both increases AP-induced EPSP suppression and occludes the impact of apamin. To test this idea, we investigated AP-induced EPSP suppression in an active model of a layer 5 pyramidal neuron (see

Methods). This model included a morphologically realistic basal dendrite, the primary site of synaptic input from surrounding layer 5 pyramidal neurons (*Markram et al., 1997a*), with explicitly modelled dendritic spines (*Figure 5a*). The properties of the basal dendritic tree were taken from previously published work (*Nevian et al., 2007*), and showed AP backpropagation similar to that observed experimentally (*Nevian et al., 2007*). SK channels were constrained to dendritic spines, uniformly distributed, and activated solely by calcium influx through voltage-activated calcium channels (also constrained to spines), consistent with the observation that in cortical pyramidal neurons NMDAR activation during EPSPs does not drive SK channel activation (*Figure 1c* and *Figure 2e*).

EPSPs containing both AMPA and NMDA components were generated in a small subset of spines and located at intermediate basal dendritic locations (130–170 μm from the soma), similar to the location expected for synaptic connections between layer 5 pyramidal neurons (*Markram et al., 1997a*). Using this model all aspects of the experimental data could be reproduced, including SK-dependent EPSP suppression by preceding APs (*Figure 5b*; compare with *Figure 1b*), as well as the observation that EPSP suppression was enhanced following NMDAR block (*Figure 5b,c*; compare with *Figure 4a*), and the finding that NMDAR block removed the dependence of EPSP suppression on SK channels (*Figure 5c*; compare with *Figure 4a*). In addition, as observed experimentally, SK-dependent EPSP suppression in the model was dependent on AP-EPSP timing, with a timing of −10 ms suppressing EPSP peak by a similar amount to that seen experimentally (*Figure 5d*; compare with *Figure 1e*). SK-dependent EPSP suppression by preceding APs also decreased EPSP width (*Figure 5e*; compare with *Figure 1f*), and was greater for more distally generated EPSPs (*Figure 5f*; compare with *Figure 1g*).

We used the model to investigate the impact of SK channels on spine voltage during backpropagating APs. These simulations indicated that switching off SK channels in spines increases backpropagating AP width, but not amplitude, with the greatest effect at distal dendritic locations (*Figure 5g–i*). This effect of SK channels leads to depolarisation of spine voltage as well as an increase in spine calcium influx through voltage-activated calcium channels (*Figure 5j–l*), which is greater for more distal spines and matches recent experimental observations (*Jones and Stuart, 2013*). Together, these simulations suggest that SK channels in spines act to repolarize backpropagating APs and have a greater impact at more distal dendritic locations.

Using this model, we investigated the mechanisms underlying EPSP suppression by APs and its dependence on SK channels. As observed in our experiments, preceding APs slightly reduced aEPSPs generated by somatic current injection in an SK-independent manner due to conductances activated during the ADP (*Figure 6a*). As SK channels in the model were constrained to dendritic spines these simulations show that activation of dendritic SK channels by backpropagating APs has minimal impact on somatic EPSPs, supporting the conclusion from the aEPSP experiments that the impact of SK channels on unitary EPSPs is dendritic. We next used the model to determine the changes in AMPA and NMDA synaptic current when APs are evoked before EPSPs (−10 ms timing). Preceding APs had opposing effects on current flow through AMPA and NMDA channels, leading to a decrease in AMPA current due to a reduction in driving force (*Figure 6b*), but an increase in NMDA current due to the impact of spine depolarisation during backpropagating APs on NMDA channel activation (*Figure 6c*). Switching SK channels off in these simulations had minimal impact on AMPA current flow (*Figure 6b*), but further increased NMDA current due to increased depolarisation of dendritic spines following broadening of backpropagating APs (*Figure 6c*; see *Figure 5j*). Switching off SK channels also led to an increase in NMDA current during pairing of EPSPs with APs at positive timings (+10 ms; not shown). In conclusion, these stimulations indicate that preceding APs lead to EPSP suppression by reducing current flow through AMPA receptors, which is offset by NMDAR activation (*Figure 6d*). As the extent of NMDAR activation is controlled by SK channels, blocking SK channels reduces EPSP suppression, whereas blocking NMDARs increases EPSP suppression.

We next investigated whether SK-dependent EPSP suppression by APs occurs locally in activated spines or requires global activation of SK channels. In simulations where SK channels were restricted to active spines receiving synaptic input (the vast minority), turning SK channels off had essentially no impact on AP-induced EPSP suppression (*Figure 6e*). In contrast, when SK channels were located in inactivated spines that did not receive synaptic input (the vast majority), blocking SK channels reduced EPSP suppression to a similar extent to that seen when SK channels were located in all spines (*Figure 6f*; compare with *Figure 5b*). These simulations indicate that SK-dependent suppression of EPSPs by APs is not due to the local impact of SK channels activated solely in spines receiving

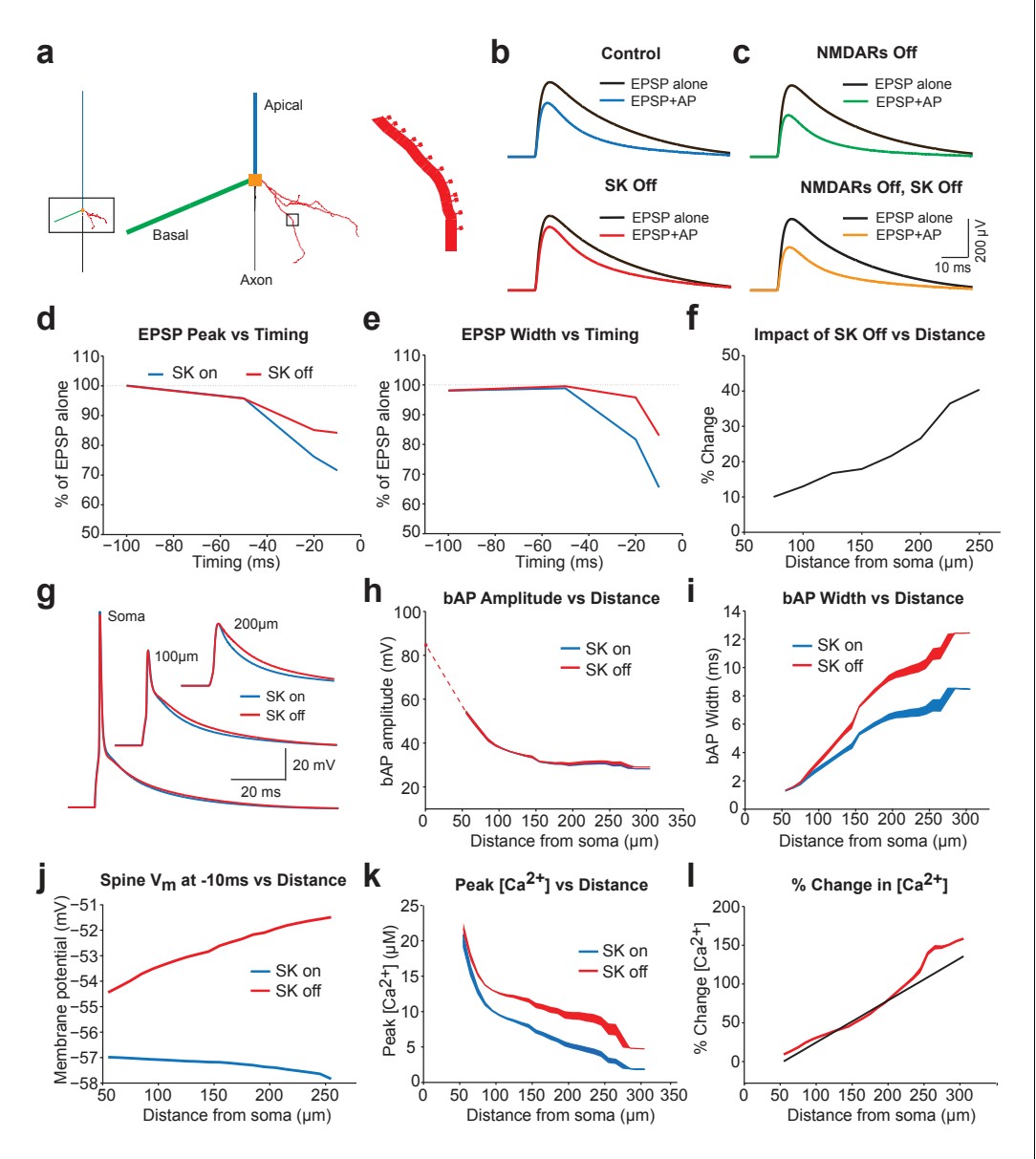

**Figure 5.** Modelling EPSP suppression by APs. (**a**) Left, Morphology of a reduced layer 5 pyramidal neuron model drawn to scale. Middle and right, Magnified images of the boxed regions showing basal dendrite with spines (red). (**b**) Somatic EPSPs evoked alone (black) and 10 ms after APs (blue and red) in control (top) and with SK channels off (bottom). (**c**) Somatic EPSPs evoked alone (black) and 10 ms after APs (green and yellow) with NMDARs off (top) and both NMDAR and SK channels off (bottom). (**d,e**) Somatic EPSP amplitude (**d**) and width (**e**) for EPSPs evoked at different times after APs relative to the amplitude of EPSPs alone in control (blue) and with SK channels off (red). (**f**) Percentage increase in somatic EPSP amplitude after switch off of SK channels for EPSPs evoked 10 ms after APs versus synapse location. (**g**) APs at the soma and in spines at the indicated basal dendritic locations in control (blue) and with SK channels off (red). (**h,i**) Backpropagating AP (bAP) amplitude (**h**) and width (**i**) in spines versus distance from the soma. The dotted line indicates dendritic locations without spines. (**j**) Membrane potential 10 ms after AP onset in spines versus distances from the soma in control (blue) and with SK channels off (red). (**k,l**) Peak (**k**) and percentage change (**l**) in spine calcium concentration during backpropagating APs versus distance from the soma in control (blue) and with SK channels off (red). Line (**l**) represents fit to experimental data (*Jones and Stuart, 2013*).

DOI: https://doi.org/10.7554/eLife.30333.006

synaptic input, but requires global recruitment of SK channels along the entire length of the active

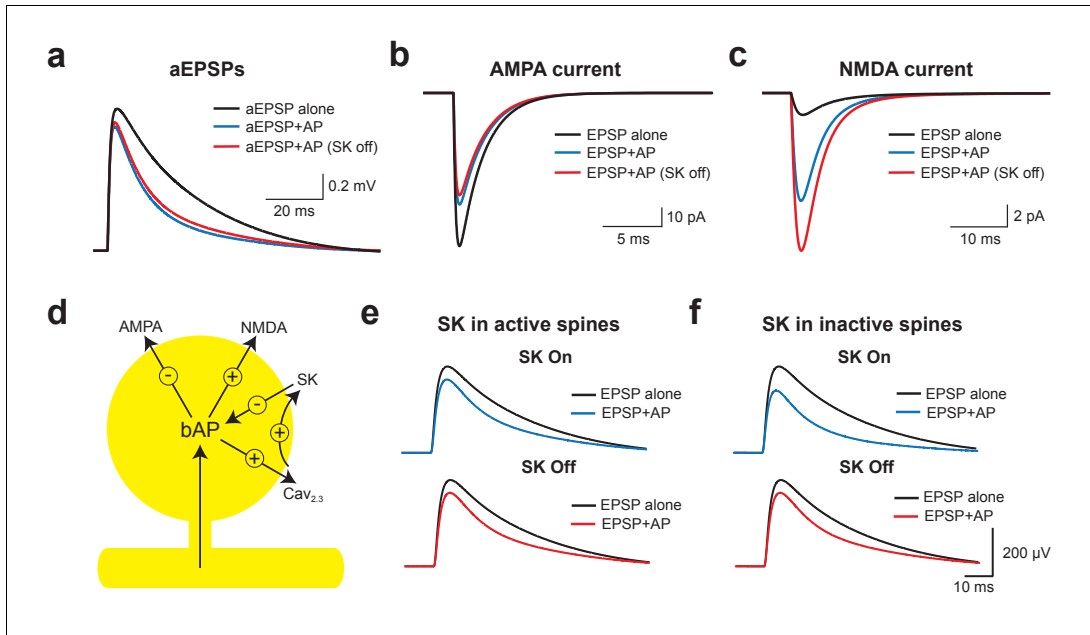

**Figure 6.** Mechanisms of EPSP suppression by APs. (a) Somatic artificial EPSPs generated by somatic current injection alone (black) and 10 ms after APs in control (blue) and with SK channels off (red). (b,c) AMPA (b) and NMDA current (c) in spines receiving synaptic input during EPSPs alone (black) and 10 ms after APs in control (blue) and with SK channels off (red). (d) Schematic of the positive and negative impact of backpropagating APs (bAPs) on current flow through AMPA, NMDA, SK and voltage-activated calcium channels in spines. (e,f) Somatic EPSPs alone (black) and 10 ms after APs (blue and red) in control (top) and with SK channels off (bottom) in simulations were SK channels were located only in active spines receiving synaptic input (e) or only in inactive spines which did not receive synaptic input (f).
DOI: https://doi.org/10.7554/eLife.30333.007

basal dendrite.

## SK channels regulate spike-timing dependent synaptic plasticity

As the extent of NMDAR activation is crucial for determining the magnitude and sign of STDP (*Bi and Poo, 1998*; *Kampa et al., 2006*; *Markram et al., 1997b*; *Sjöström et al., 2001*), activation of SK channels by backpropagating APs is likely to influence the induction of STDP during AP-EPSP pairing. To test this idea, we induced STDP during recordings from synaptically connected layer 5 pyramidal neurons using single AP-EPSP pairing (60 repetitions at 0.2 Hz) at −10 ms and +10 ms in the absence and presence of apamin. When bi-directional connections were obtained, the impact of AP-EPSP pairing at −10 ms and +10 ms was tested simultaneously. Consistent with earlier work in layer 5 pyramidal neurons in somatosensory cortex (*Kampa et al., 2006*; *Markram et al., 1997b*), low-frequency (0.2 Hz) single AP-EPSP pairing at both positive (+10 ms) and negative times (−10 ms) did not lead to STDP (*Figure 7a–d*). In contrast, following block of SK channels with apamin single AP pairing at −10 ms induced robust LTD, whereas pairing at +10 ms induced robust LTP (*Figure 7a–d*; n = 6 and n = 5, respectively). These data indicate that dendritic SK channels play a key role in regulating the induction of STDP in layer 5 pyramidal neurons in somatosensory cortex during single AP-EPSP pairing, presumably through their capacity to limit NMDAR activation. Finally, given that EPSP suppression by backpropagating APs is greater at EPSPs located at distal dendritic locations (*Figure 1g*), we investigate the dependence of STDP observed after blocking SK channels on EPSP location. This analysis indicated that EPSPs with slower rise times, which are presumably generated at more distal locations, underwent greater plasticity in the presence of apamin during pairing at both positive and negative times (*Figure 7e*).

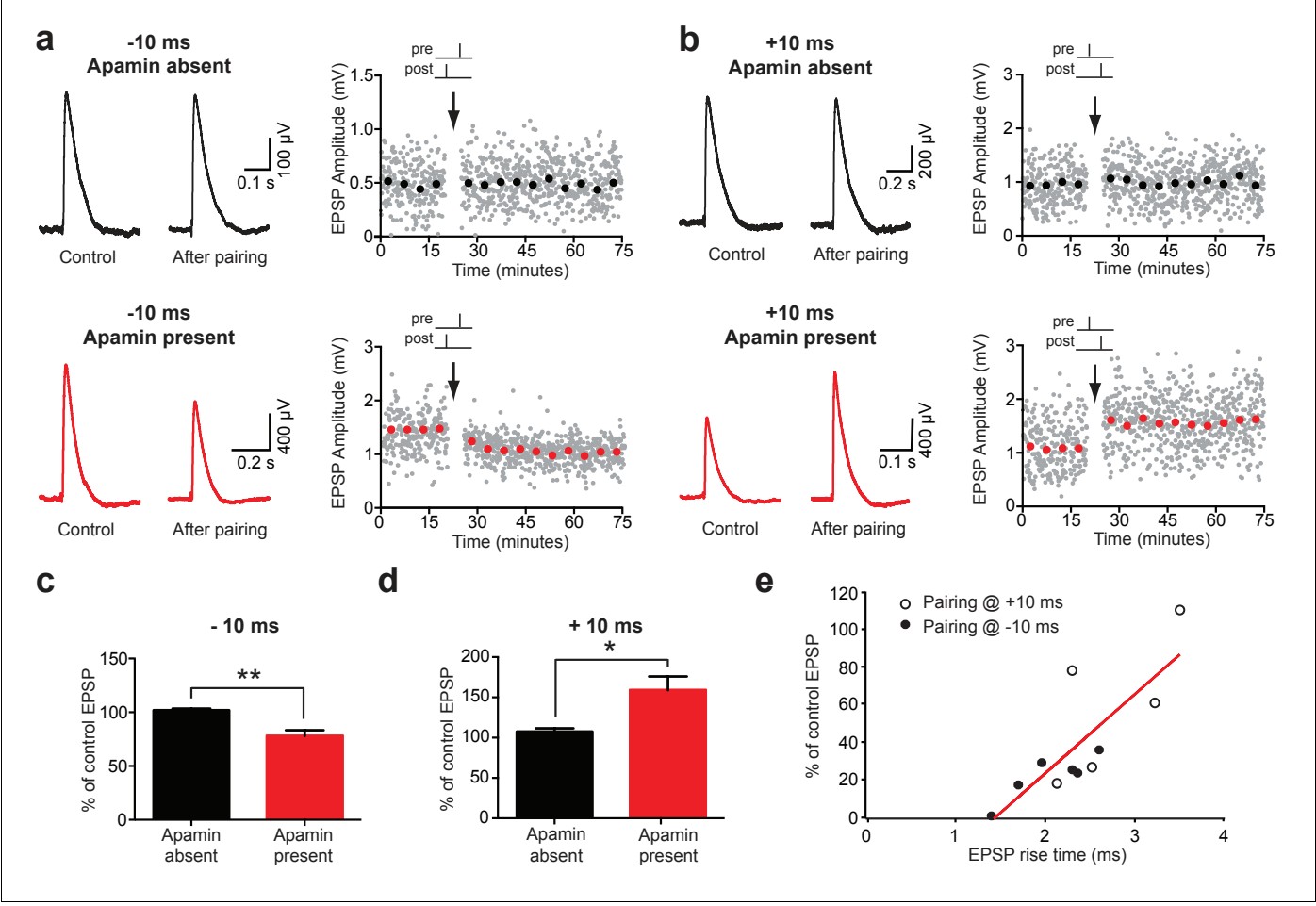

**Figure 7.** Impact of SK channels on spike-timing dependent synaptic plasticity. (**a,b**) Left, EPSPs before and after AP-EPSP pairing at −10 ms (**a**) and + 10 ms (**b**) in control (top, black) and apamin (bottom, red). Right, Plot of EPSP amplitude versus time. Each point represents the average response over 5 min. (**c,d**) Average % change in EPSP amplitude 20–30 min after AP-EPSP pairing relative to baseline during AP-EPSP pairing at −10 ms (**c**; n = 7) and + 10 ms (**d**; n = 5) in control (black) and apamin (red). (**e**) Plot of the absolute % change in EPSP amplitude in apamin 20–30 min after AP-EPSP pairing at −10 ms (filled circles) and + 10 ms (open circles) versus EPSP rise time. Data fitted with a linear regression (r = 0.8; p<0.01).

DOI: https://doi.org/10.7554/eLife.30333.008

## Discussion

In this study, we show that backpropagating APs suppress EPSPs in both cortical and hippocampal pyramidal neurons if generated within a brief time window before EPSP onset. In layer 5 pyramidal neurons, EPSP suppression by backpropagating APs was dependent on the distance of activated synapses from the soma, was dendritic in origin, and involved SK-dependent suppression of NMDA receptor activation. Together, these effects of SK channel activation gate STDP induction during low-frequency AP-EPSP pairing, with both LTP and LTD absent under control conditions but present after SK channel block.

Previous work in both cortical pyramidal neurons as well as cerebellar Purkinje neurons have described an inhibitory action of postsynaptic APs on EPSPs, due to different mechanisms from those described here. APs evoked after EPSP onset have been shown to shunt EPSPs at the soma of cortical layer 5 pyramidal neurons and cerebellar Purkinje neurons, reducing their amplitude (*Häusser et al., 2001*). This effect is due to conductances near the soma activated during the AP itself, and is only observed when APs are generated after EPSP onset. It is therefore quite different from the EPSP suppression described here, which is dendritic in origin, requires SK channel activation and is observed when the AP occurs before EPSP onset.

More relevant to our observations is the work by *Froemke et al. (2005)*, who showed that preceding APs lead to EPSP suppression in cortical layer 2/3 pyramidal neurons. This effect was generated when postsynaptic APs were evoked before EPSP onset, and had a similar time window to the EPSP suppression described here in layer 5 pyramidal neurons. In contrast to our findings, however, this study attributed EPSP suppression to calcium-dependent NMDAR desensitisation, due to its dependence on NMDAR activation and intracellular calcium (*Froemke et al., 2005*). As we show SK-dependent EPSP suppression by preceding APs In layer 5 pyramidal neurons is also dependent on NMDAR activation and presumably intracellular calcium, given the requirement for SK channel activation. However, in contrast to the observations of *Froemke et al. (2005)*, we find that EPSP suppression by preceding APs in both layer 2/3 and layer 5 cortical pyramidal neurons was largely blocked by apamin. Based on these observations, we predict that the EPSP suppression described by *Froemke et al. (2005)* is likely to be mediated by the same SK-dependent mechanism described here, rather than via calcium-dependent NMDAR desensitisation.

Previous work indicates that SK channels in hippocampal CA1 neurons can be activated by EPSPs alone, regulating NMDAR activation and EPSP amplitude (*Bloodgood and Sabatini, 2007*; *Ngo-Anh et al., 2005*; *Wang et al., 2014*). Consistent with these earlier observations, we also observed that blocking SK channels increased EPSPs evoked on their own in CA1 pyramidal neurons. Interestingly, this does not occur in cortical layer 2/3 or layer 5 cortical pyramidal neurons, despite the presence of SK channels in spines on basal dendrites of layer 5 pyramidal neurons (*Jones and Stuart, 2013*). The reason for this difference is unclear, but may involve differences in the properties or level of expression of NMDARs and/or SK channels in the cortex compared to the hippocampus. Alternatively, differences in the site of synaptic input onto pyramidal neurons in these brain regions may lead to different extents of NMDAR activation. Synaptic connections between layer 5 neurons are dispersed onto multiple basal dendritic branches (*Markram et al., 1997a*), reducing the voltage change at individual inputs. In contrast, synapses in other brain regions may cluster, leading to more localised dendritic depolarisation and enhanced NMDAR activation. It is also possible that the calcium source for SK channel activation is different in cortical compared to hippocampal CA1 pyramidal neurons. Activation of SK channels in spines on basal dendrites of layer 5 pyramidal during backpropagating APs has recently been shown to be dependent on only R-type voltage-dependent calcium channels (*Jones and Stuart, 2013*). In contrast, SK channels in hippocampal CA1 pyramidal neurons are thought to be activated by calcium influx through NMDARs (*Ngo-Anh et al., 2005*; *Wang et al., 2014*; although see *Bloodgood and Sabatini, 2007*).

The SK-dependent component of EPSP suppression by preceding APs was absent when NMDARs were blocked, indicating a critical role of NMDARs. Surprisingly, these experiments revealed that blocking NMDARs increased EPSP suppression. This increase in AP-induced EPSP suppression arises because under control conditions the extent of EPSP suppression due to reduced current flow through AMPA receptors is offset by increased NMDAR activation. As a result blocking NMDARs increases EPSP suppression. In addition, we observed that following NMDAR block SK channels had no impact on EPSP suppression. This is presumably the case as SK channels predominately modulate NMDAR activation, and have little impact on current flow through AMPA receptors (see *Figure 6b, c*). In the absence of NMDARs blocking SK channels therefore has little impact on EPSP suppression.

In summary, these data suggest that NMDAR activation is enhanced by preceding APs, with the extent of NMDAR activation kept in check by dendritic SK channels. Blocking NMDAR activation therefore both increases EPSP suppression, and removes its dependence on SK channels. This idea was supported by our model. These simulations showed that backpropagating APs lead to significant depolarisation of dendritic spines on basal dendrites, with the extend of spine depolarisation controlled by SK channel activation. This effect of SK channels on spine voltage presumably also underlies their capacity to regulate spine calcium influx during backpropagating APs (*Jones and Stuart, 2013*). Depolarisation of dendritic spines by backpropagating APs not surprisingly increases NMDAR activation due to relief of magnesium block (*Kampa et al., 2006*; *Mayer et al., 1984*; *Nowak et al., 1984*), while at the same time reduces current flow through AMPA receptors due to a reduction in electrochemical driving force. As SK channels control spine voltage during backpropagating APs they regulate the extent of NMDAR activation, which presumably also explains their capacity to gate STDP induction.

The model also revealed that EPSP suppression was not due to SK channel activation localised solely to active spines, but required SK channel activation along the entire dendritic branch. As a

result the impact of SK channel activation on backpropagating AP width and thereby spine depolarization is greater at more distal dendritic locations. This can explain why the extent of AP-induced EPSP suppression was greater for more distal synaptic inputs (*Figure 1g*), and may also explain the great impact of preceding APs on distal compared to proximal EPSPs in earlier work in layer 2/3 neurons (*Froemke et al., 2005*). This dependence of EPSP suppression on synapse location also impacted on STDP, with the absolute magnitude of STDP observed during AP-EPSP pairing at both positive and negative times positively correlated with somatic EPSP rise time (*Figure 7e*).

STDP is a form of synaptic plasticity where the strength of synaptic connections is modified by the precise timing of synaptic input and postsynaptic APs (*Bi and Poo, 1998*; *Markram et al., 1997b*; *Sjöström et al., 2001*). While there are many demonstrations of LTP induced by low-frequency pairing of single pre- and postsynaptic APs (*Bi and Poo, 1998*; *Dan and Poo, 2004*; *Froemke et al., 2005*; *Nevian and Sakmann, 2006*), in some neuronal cell types more intense stimuli such as high AP frequency bursts are required for plasticity to occur. For example, unitary synaptic connections between layer 5 pyramidal neurons do not display LTP when EPSPs are paired with APs at positive timings (+10 ms) in both somatosensory and visual cortex (*Kampa and Stuart, 2006*; *Markram et al., 1997b*; *Sjöström et al., 2001*). Multiple pre- and postsynaptic AP pairings at high frequencies (10–40 Hz) or pairing single presynaptic APs with additional postsynaptic depolarisation or a postsynaptic AP burst is required (*Kampa et al., 2006*; *Markram et al., 1997b*; *Sjöström and Häusser, 2006*; *Sjöström et al., 2001*). Low-frequency single AP pairing of unitary EPSPs between layer 5 pyramidal neurons also does not produce LTD in somatosensory cortex (*Kampa et al., 2006*; *Markram et al., 1997b*), although this does lead to LTD in visual cortex (*Sjöström et al., 2001*). These data suggest that the depolarisation associated with backpropagating APs is in some cases insufficient for STDP induction, which has lead some to question the physiological relevance of STDP (*Lisman and Spruston, 2005*).

As we show here, activation of SK channels by backpropagating APs gates the induction of both LTP and LTD during low-frequency single AP-EPSP pairing at unitary connections between layer 5 pyramidal neurons in somatosensory cortex, presumably by restricting NMDAR activation. Why blocking SK channels is not required for LTD induction at unitary connections between layer 5 pyramidal neurons in visual cortex is unclear, but may have to do with region-specific differences in input resistance. The input resistance of layer 5 pyramidal neurons is significantly higher in visual compared to somatosensory cortex (*Sjöström et al., 2001*; *Williams and Stuart, 1999*). As a result, under control conditions backpropagating APs may lead to sufficient depolarization of dendritic spines to cause LTD induction in visual cortex layer 5 pyramidal neurons even in the presence of SK channel activation. Alternatively, regional differences in SK channels expression or activation may be involved, as presumably underlie the different impact of SK channels on EPSPs evoked on their own in cortical compared to hippocampal CA1 pyramidal neurons.

The capacity of SK channels to gate the induction of STDP during low-frequency single AP-EPSP pairing provides a means to negate the impact of accidental or spurious AP-EPSP pairings on synaptic plasticity. In addition, it provides a mechanism to regulate the extent of STDP induction given that the availability SK channels is under the control of both metabotropic acetylcholine and glutamate receptors (*Buchanan et al., 2010*; *Giessel and Sabatini, 2010*; *Tigaret et al., 2016*), as well as plasticity itself (*Lin et al., 2008*).

When APs are evoked shortly after EPSP onset (e.g. +10 ms), while EPSPs at the soma will be shunted, as previously described (*Häusser et al., 2001*), at the level of individual spines the depolarization associated with backpropagating APs will increase NMDAR activation and associated calcium influx (*Koester and Sakmann, 1998*; *Nevian and Sakmann, 2006*; *Schiller et al., 1998*; *Yuste and Denk, 1995*), which would be expected to be regulated by SK channel channels, as occurs when APs are evoked before ESPs. Consistent with this idea, simulations using our model indicated that when APs were generated 10 ms after EPSP onset turning SK channels off in the model increased NMDAR activation, while having essentially no impact on AMPA current (not shown). This effect presumably underlies the capacity of SK channels to gate STDP at both positive and negative spike timings.

In summary, we find that backpropagating APs have opposing effects on the currents generated during EPSPs. APs evoked just before EPSP onset reduce the electrochemical driving force for current flow through AMPA receptors, leading to EPSP suppression. At the same time, the depolarisation associated with backpropagating APs increases NMDAR activation, opposing the reduction in AMPA current. As the extent of NMDAR activation is controlled by dendritic SK channels, blocking

these channels increases NMDAR activation, reducing AP-induced EPSP suppression and leading to STDP induction.

## Materials and methods

### Preparation

Wistar rats (16–18 days old) of either sex were anaesthetized by inhalation of isofluorane and decapitated according to guidelines approved by the Animal Experimentation Ethics Committee of the Australian National University. Parasagittal brain slices of somatosensory cortex (300 µm), or transverse slices of hippocampus (300 µm), were made using a Leica VT1000S tissue slicer (Germany). During slicing the brain was submerged in ice-cold cutting artificial cerebrospinal fluid (ACSF) containing: 125 mM NaCl, 25 mM NaHCO$_3$, 15 mM sucrose, 10 mM glucose, 3 mM KCl, 1.25 mM NaH$_2$PO$_4$, 0.5 mM CaCl$_2$, 7 mM MgCl$_2$, 4 mM ascorbic acid, 2 mM Na-Pyruvate, bubbled with 95% O$_2$/5%CO$_2$ (pH 7.4). Slices were then transferred to a holding chamber and incubated at 35°C for 10–15 min in the cutting solution before being transferred to another chamber containing standard ACSF: 125 mM NaCl, 25 mM NaHCO$_3$, 25 mM glucose, 3 mM KCl, 1.25 mM NaH$_2$PO$_4$, 2 mM CaCl$_2$, 1 mM MgCl$_2$ bubbled with 95%O$_2$/5%CO$_2$ (pH 7.4) and maintained at room temperature until the commencement of experiments (1–2 hr later). During recording slices were perfused at a rate of ~4 mL/min with standard ASCF maintained at 35–36°C.

### Electrophysiology

Somatic whole-cell current-clamp recordings were made from visually identified layer 5, layer 2/3 or hippocampal CA1 pyramidal neurons using differential interference contrast (DIC) optics (*Stuart et al., 1993*). Recording electrodes (3–4 MΩ) were filled with an intracellular solution containing: 130 mM K-Gluconate, 10 mM HEPES, 10 mM KCl, 4 mM Mg$_2$ATP, 0.3 mM Na$_2$GTP and 10 mM Na$_2$-Phosphocreatine (pH 7.2 with KOH). In some experiments, 1,2-bis(o-aminophenoxy)ethane-N,N, N',N'-tetraacetic acid (BAPTA) was added to the intracellular solution at a concentration of 300 µM, as denoted in the Results. Recordings were made using identical BVC 700A amplifiers in current-clamp mode (Dagan, MN). Series resistance was monitored throughout the experiment and was typically below 10 MΩ. Voltage traces were filtered at 10 kHz and digitized at 50 kHz using an ITC-18 computer interface (HEKA, Germany). Data acquisition was performed using AxoGraph X software (AxoGraph Scientific, Australia) running on an Apple iMac computer.

Data were only analyzed from cells with a stable resting membrane potential (less than 2 mV change) more negative than −60 mV, and a stable input resistance (less than 10% change; measured using current injections of −50 pA of 800 ms duration). All recordings were made at the resting membrane potential in the absence of injected current. During paired recordings presynaptic APs were generated by brief (2 ms, 3 nA) somatic current pulses at a frequency of either 0.1 or 0.2 Hz. Data analysis was restricted to recordings where the unitary EPSP amplitude was 1 mV or less. For experiments on layer 2/3 and hippocampal CA1 pyramidal neurons extracellular stimulation was used to evoke EPSPs using a custom-built theta glass bipolar extracellular stimulating electrode. In these experiments Alexa Fluor 488 (Invitrogen) was dialysed into the neuron and Alexa 555 (Invitrogen, CA) was added to the stimulating electrode solution. This allowed both the dendritic tree of the neuron and the tip of the stimulating electrode to be visualized using fluorescence microscopy. The tip of the stimulating electrode was placed within 10–15 µm of the dendrite (layer 2/3: basal; CA1: oblique) of the neuron of interest. EPSPs were generated at 0.1 Hz using a short negative current pulse (10–20 µA, 0.2 ms) from an isolated stimulator (World Precision Instruments, FL). The stimulating electrode was filled with 1M NaCl as this was found to significantly reduce the current required to evoke the desired postsynaptic response and thus reduced the stimulus artefact. The stimulus current was set to produce an EPSP of approximately 2 mV.

To investigate the impact of APs on EPSPs, EPSPs were evoked alone or together with preceding postsynaptic APs evoked by brief (2 ms, 3 nA) somatic current pulses (10 ms before EPSP onset, unless otherwise stated). Postsynaptic APs generated on their own were interleaved between trials and digitally subtracted from the AP-EPSP waveform to reveal the impact of APs on EPSPs. For experiments performed with BAPTA added to the intracellular solution, whole-cell recordings were initially established without BAPTA added to the intracellular solution. After a connection was

identified, the patch pipette on the postsynaptic neuron was withdrawn and the neuron repatched with a pipette containing BAPTA in the intracellular solution, which was then allowed to dialyse into the neuron for 30 min before experiments commenced. For STDP experiments, EPSPs were evoked at 0.2 Hz (every 5 s) and plasticity induced by 60 repetitions of single pre- and postsynaptic APs at a time interval of +10 or −10 ms at 0.2 Hz.

## Artificial voltage waveforms

Somatic current injection with an exponential rise and decay of 0.3 ms and 3 ms, respectively, was used to generate somatic artificial EPSPs (aEPSPs). These kinetics were chosen to mimic spontaneous EPSCs measured at the soma of layer 5 neurons (*Stuart and Sakmann, 1995*).

## Pharmacology

SK channels were blocked by apamin (100 nM; Sigma Aldrich, Australia) and NMDARs blocked by D (-)−2-amino-5-phosphono-pentanoic acid (AP5; 100 µM; Tocris Bioscience, UK). Both agents were bath applied. Steady-state concentrations in the recording chamber were achieved within 5 min after the commencement of drug wash in, and the impact evaluated after 20–30 min.

## Data analysis

All data were analysed using Axograph X. Statistical significance between groups was evaluated with T-tests in either the paired or unpaired configuration, as appropriate. Statistics were performed in Graphpad Prism 6. In the figures a single asterisk (*) indicates $p<0.05$, double asterisks (**) indicates $p<0.01$, and 'n.s.' indicates not statistically significant ($p>0.05$).

## Neuronal modelling

Computer simulations were performed using the NEURON 7.2 simulation environment on a Linux desktop computer running Ubuntu 12.04 LTS. A reduced layer 5 pyramidal neuron model that displays realistic AP kinetics and firing was used to reduce computational time (*Bahl et al., 2012*). A fully reconstructed basal dendritic branch obtained from a morphologically realistic layer 5 pyramidal neuron model (*Stuart and Spruston, 1998*) was attached to the soma of this model. Specific membrane resistance ($R_m$; 15,000 Ω.cm$^2$), capacitance ($C_m$; 1 µF/cm$^2$) and internal resistance ($R_i$; 105 Ω.cm) were uniformly distributed throughout the reconstructed basal dendritic branch based on previously published values (*Nevian et al., 2007*). A total of 365 spines were explicitly modeled and placed at ~2 µm intervals along the reconstructed basal dendrite, starting 50 µm from the soma. Each spine consisted of a head and neck segment. Spine heads had diameters and lengths of 500 nm. Spine necks were 1 µm long and had diameters of 160 nm, producing a spine neck resistance of approximately 50 MΩ. In addition to adding spines explicitly, $R_m$ was decreased and $C_m$ increased by a factor of 1.73 in dendritic compartments greater than 50 µm from the soma of the reconstructed basal dendritic branch to account for the impact of spines not explicitly modeled on membrane properties. This correction adjusts for the fact than only ~25% of the expected number of spines on the reconstructed basal dendritic branch were explicitly modeled, and assumes that spines effectively double the surface area of dendritic compartments in which they are present. The resting membrane potential of the model was set to −79.4 mV.

Voltage-activated calcium channels (*Aradi and Holmes, 1999*) were included in spine heads at the same density at all dendritic locations, with the density scaled to give a free calcium concentration of 20 µM in proximal spines during backpropagating APs, as observed experimentally (*Cornelisse et al., 2007*; *Sabatini et al., 2002*). The spine calcium concentration in individual spines was evaluated by summing the calcium currents through all voltage-gated calcium channels in the spine head. Calcium removal from the spine head was modeled by a single exponential decay with a time constant of 9 ms, similar to experimental observations (*Sabatini et al., 2002*). The calcium-dependence of SK channel model (*Somjen et al., 2008*) used a Hill function:

$$\frac{\text{gskbar}}{1+\left(\frac{K_D}{Ca_i}\right)^{\text{Hill}}}$$

where *gskbar* is the SK conductance, $Ca_i$ is the intracellular calcium concentration, *Hill* is the Hill

coefficient (4.7) and $K_D$ is the calcium dissociation constant (300 nM see, *Xia et al., 1998*). SK channels were only included in spine heads at a uniform density of 20 pS/µm². This value was chosen to replicate experimental data on the impact of SK channels on calcium influx into spines during back-propagating APs (*Figure 5l*; see *Jones and Stuart, 2013*). It over-estimates the required SK conductance per spine as only ~25% of the spines in the reconstructed dendrite were explicitly modeled. As we observed no impact of somatic SK channels on EPSP amplitude, no SK channels were included at the soma. Also, as our experimental observations indicated that SK channels in cortical pyramidal neurons are not activated by NMDARs, SK channel activation was coupled solely to calcium flux through voltage-dependence calcium channels. No other voltage or calcium-activated conductances were included in the reconstructed basal dendrite.

Excitatory synapses were made onto spines on the reconstructed basal dendritic branch at different locations, as indicated in the Results. At each location two synapses within a selected region were randomly chosen. The conductance change at these synapses had both AMPA and NMDA components. The AMPA conductance had exponential rise and decay of 0.2 and 2 ms, respectively, a peak of 500 pS and a reversal potential of 0 mV. The NMDA conductance (2.9 nS) was based on a published model (*Moradi et al., 2013*) with modified kinetics and voltage dependence. It had an exponential rise time (1 ms) and double exponential decay (5 and 30 ms). The voltage dependence of the NMDAR model was described by:

$$\mathrm{Mg(V)} = \frac{1}{1 + 0.0501 \mathrm{e}^{(-0.105\mathrm{V})}}$$

The NMDAR model had a $V_{1/2}$ of $-28.5$ mV and a slope of 0.0262 mV$^{-1}$, and has similar voltage dependence to a kinetic model developed from experimental data in layer 5 pyramidal neurons (*Kampa et al., 2004*). The amplitude, kinetics and voltage-dependence of the NMDAR model were tuned to reproduce the impact of preceding APs on EPSP peak and width, as well as the increase in EPSP suppression observed after NMDAR block.

## Acknowledgements

The authors wish to thank Dr. Tobias Bock for assistance with MATLAB programming. This work was supported by the National Health and Medical Research Council of Australia, the Australian Research Council and The John Curtin School of Medical Research.

## Additional information

### Funding

| Funder | Author |
| --- | --- |
| Australian Research Council | Greg J Stuart |
| National Health and Medical Research Council | Greg J Stuart |
| The John Curtin School of Medical Research | Greg J Stuart |

The funders had no role in study design, data collection and interpretation, or the decision to submit the work for publication.

### Author contributions

Scott L Jones, Conceptualization, Data curation, Formal analysis, Investigation, Writing—original draft, Writing—review and editing; Minh-Son To, Software, Formal analysis, Validation, Investigation, Writing—review and editing; Greg J Stuart, Conceptualization, Resources, Supervision, Funding acquisition, Writing—original draft, Project administration, Writing—review and editing

### Author ORCIDs

Greg J Stuart http://orcid.org/0000-0001-9395-2219

### Ethics

Animal experimentation: This research was performed according to guidelines approved by the Animal Experimentation Ethics Committee of the Australian National University (Animal ethics protocol number A2014_63).

### Decision letter and Author response

Decision letter https://doi.org/10.7554/eLife.30333.010
Author response https://doi.org/10.7554/eLife.30333.011

## Additional files

### Supplementary files

• Transparent reporting form
DOI: https://doi.org/10.7554/eLife.30333.009

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
