## [Decision Letter]

Thank you for submitting your article "Dendritic SK channels activated by action potentials suppress EPSPs and gate spike-timing dependent synaptic plasticity" for consideration by *eLife*. Your article has been favorably evaluated by Gary Westbrook (Senior Editor) and three reviewers, one of whom is a member of our Board of Reviewing Editors. The following individual involved in review of your submission has agreed to reveal their identity: Stefan Remy (Reviewer #2).

The reviewers have discussed the reviews with one another and the Reviewing Editor has drafted this decision to help you prepare a revised submission.

Summary:

In this manuscript the authors applied paired whole-cell recordings of cortical and hippocampal pyramidal neurons in combination with computational modeling to investigate the role of small conductance calcium-activated potassium channels (SK channels) in the regulation of EPSP amplitudes by back-propagating action potentials (APs) and their potential effects on spike-timing dependent synaptic plasticity (STDP). The main finding is that back-propagating APs reduced EPSP size in a distance-dependent manner. The suppression was more effective with higher distances from the soma and was mediated by dendritic – and not somatic – SK channel activation. One important requirement for this SK-mediated effect is, that back-propagating APs are evoked ~10 ms prior to EPSP induction. Pharmacological block of SK channels increased NMDAR activation which in turn reduced AP-mediated EPSP suppression and led to the expression of long-lasting synaptic plasticity. Moreover, SK channel activation served to suppress the induction of spike timing-dependent plasticity during low-frequency single-AP-EPSP pairings. In addition, the authors report potentially important functional differences between cortical and hippocampal pyramidal neurons. This is an important study on the role of the neurons' excitability on EPSP amplitude and synaptic plasticity on the basis of the interaction between SK channels, NMDARs and back-propagating APs. It is a technically well performed and well written study. However, the reviewers have some main concerns which are summarized below:

Essential revisions:

1) Please clarify the wording 'distance-dependent'. On the basis of previous work of the authors and the rise times of EPSPs presented in this study, the authors should be able to give more precise statements on distance between dendritic EPSP induction and the somatic recording site. The morphology of the cells and synapse location (in case intracellular labelling is available) should be helpful in this context.

2) Please give more explanations in the Discussion on the nature of the potential influence of perisomatic conductances on the shape of EPSPs. The authors propose that the SK-insensitive component of EPSP suppression is somatic in origin and very likely caused by conductance's related to spike after hyperpolarization. The argument for this conclusion is only indirect and based on similar sizes in EPSP amplitude modulation, but not shown directly by for example blocking the potentially underlying conductances. Either provide additional explanations on which conductances underlie EPSP attenuation or tone down this conclusion.

3) Please discuss whether non-linear specific membrane resistances and gradients in dendritic SK channel densities may also be factors influencing the shape of dendritically evoked EPSP.

4) Please make any code you have developed for data analysis as well as modelling and which is essential to reproduce the results of the study, publicly available.

5) Please give more explanations on the potential conductance's which may explain why in the absence of NMDAR activity action potential-induced EPSP suppression increased. Moreover, what is your explanation for why SK had no effect on action potential-induced EPSP suppression in the presence of NMDAR blockers but slightly increased the EPSP half-width? These questions require discussion.

6) The authors should compare their data with previous studies on the conditions of LTP and LTD induction using STDP protocols. In particular induction protocols from this study should be compared with previous ones, in particular from Markram et al., 1997b; Sjostrom et al., 2001 and Kampa et al., 2006. Markram et al. 1997b does show bidirectional up and down regulation of STDP but not necessarily low frequency induced plastic changes. Similarly Sjostrom et al. 2001 show that LTD was observed for 0.1 – 20 Hz. Thus, some clarification would be helpful.

7) Finally, the authors should examine the relationship of control STDP as a function of EPSP rise-time.

---

## [Author Response]

Essential revisions:1) Please clarify the wording 'distance-dependent'. On the basis of previous work of the authors and the rise times of EPSPs presented in this study, the authors should be able to give more precise statements on distance between dendritic EPSP induction and the somatic recording site. The morphology of the cells and synapse location (in case intracellular labelling is available) should be helpful in this context.

By 'distance-dependent' we mean that the magnitude of the effect is dependent on the distance from the soma. In the case of SK-dependent EPSP suppression we observed that EPSPs with longer rise times were suppressed more by preceding APs than EPSPs with shorter rise times. As longer EPSP rise times are typically associated with EPSPs generated at more distal locations we conclude that SK-dependent EPSP suppression is greater at more distal locations (that is, is “distance dependent”). We observed the same effect in our model. To avoid confusion associated with the term “distance-dependent” we have removed it from the revised manuscript and reworded the appropriate sections of the text.

In terms of the actual distance of synaptic inputs from the soma it is hard for us to be precise. Cells were not filled with biocytin to allow reconstruction and localization of synaptic contacts. The location of synaptic inputs between synaptically connected layer 5 pyramidal neurons in somatosensory cortex has been examined in detail previously by Markram et al. (1997; J. Physiol.). The range of rise times given in Markram et al. (~1 to 7 ms) is similar to that described in our study (~1.5 to 5 ms). The study of Markram et al. (1997) indicated this range was associated with synaptic connections 80 and 585 μm from the soma (mean: 147 μm; median: 105 μm), with the majority of these connections on basal dendrites. We have added this information to the revised manuscript, and directly compare our EPSP rise time data with that of Markram et al. (1997) so as to give the reader an approximate indication of the physical distance of synaptic connections from the soma.

2) Please give more explanations in the Discussion on the nature of the potential influence of perisomatic conductances on the shape of EPSPs. The authors propose that the SK-insensitive component of EPSP suppression is somatic in origin and very likely caused by conductance's related to spike after hyperpolarization. The argument for this conclusion is only indirect and based on similar sizes in EPSP amplitude modulation, but not shown directly by for example blocking the potentially underlying conductances. Either provide additional explanations on which conductances underlie EPSP attenuation or tone down this conclusion.

We agree our evidence that the SK-insensitive component of EPSP suppression is somatic is indirect. As suggested by the reviewer we have therefore toned down this conclusion in the revised manuscript.

3) Please discuss whether non-linear specific membrane resistances and gradients in dendritic SK channel densities may also be factors influencing the shape of dendritically evoked EPSP.

We did not test the role of non-linear specific membrane resistance or gradients in dendritic SK channel densities, but neither were necessary to explain our experimental observations. The SK channel density in the model was uniform. We now state this explicitly in the Materials and methods.

4) Please make any code you have developed for data analysis as well as modelling and which is essential to reproduce the results of the study, publicly available.

Once published we will make all code publicly available, as we have done for previous publications via the https://senselab.med.yale.edu/modeldb/ and http://neuromorpho.org/ websites.

5) Please give more explanations on the potential conductance's which may explain why in the absence of NMDAR activity action potential-induced EPSP suppression increased. Moreover, what is your explanation for why SK had no effect on action potential-induced EPSP suppression in the presence of NMDAR blockers but slightly increased the EPSP half-width? These questions require discussion.

The increase in AP-induced EPSP suppression observed following block of NMDARs arises because under control conditions the extent of EPSP suppression, due to reduced current flow through AMPA receptors, is offset by increased NMDAR activation. As a result blocking NMDARs increases EPSP suppression. This result was reproduced in the model and did not require conductances other than AMPA and NMDA receptors. We have reworded sections of the revised manuscript to make this point clearer.

The absence of an impact of blocking SK channels on AP-induced EPSP suppression in the presence of NMDAR blockers arises because SK channels predominately modulate NMDAR activation (see Figure 6). With regard to the slight increase in EPSP half-width observed after SK channel block in the presence of NMDAR blockers, this effect is analogous to the increase in artificial EPSP half-width observed following block of SK channels (Figure 3). We think this effect is likely to result from blocking somatic SK channels activated during the mAHP, although have no direct evidence.

6) The authors should compare their data with previous studies on the conditions of LTP and LTD induction using STDP protocols. In particular induction protocols from this study should be compared with previous ones, in particular from Markram et al., 1997b; Sjostrom et al., 2001 and Kampa et al., 2006. Markram et al. 1997b does show bidirectional up and down regulation of STDP but not necessarily low frequency induced plastic changes. Similarly Sjostrom et al. 2001 show that LTD was observed for 0.1 – 20 Hz. Thus, some clarification would be helpful.

We thank the reviewers for their comments and have reworded sections of the revised manuscript where the studies by Markram et al. (1997b), Sjostrom et al. (2001) and Kampa et al. (2006) are cited. We also thank the reviewers for calling our attention to the fact that the study by Sjostrom et al. (2001) showed that LTD can be induced during low-frequency pairing in their recordings from visual cortex. We now discuss differences between our observations and those of Sjostrom et al. (2001) explicitly. As the input resistance of layer 5 pyramidal neurons in visual cortex is significantly higher than in somatosensory cortex, dendritic spines in visual cortex are likely to be more depolarised than spines in somatosensory cortex during low-frequency pairing at negative times, despite SK channel activation. This may explain why LTD can be induction in visual cortex by low-frequency pairing at negative times, whereas this is not seen in somatosensory cortex. Alternatively, regional differences in SK channel expression or activation may be involved.

7) Finally, the authors should examine the relationship of control STDP as a function of EPSP rise-time.

We include this analysis in the revised version of the manuscript (see Figure 7). As predicted from the observation that SK channels have a greater impact on EPSPs with slower rise times, we observed that the impact of apamin on STDP was also greater for these EPSPs. We thank the reviewers for the suggestion to include this additional analysis.